



# A focal mechanism catalogue of earthquakes that occurred in the southeastern Alps and surrounding areas from 1928 – 2019

Angela Saraò, Monica Sugan, Gianni Bressan, Gianfranco Renner, Andrea Restivo[*]

[1]Istituto Nazionale di Oceanografia e Geofisica Sperimentale - OGS, Trieste, ITALY

*Correspondence to*: Angela Saraò (asarao@inogs.it)

**Abstract.** We present a focal mechanism catalogue of earthquakes that occurred in the southeastern Alps and surrounding areas from 1928 to 2019. The area involved in the process of convergence between the Adria microplate and Eurasia is one of

the most seismically active regions in the Alpine Belt. The seismicity is minor, with the Ms=6.5 Friuli earthquake being the strongest event recorded in the area, but the seismic hazard is relevant because it is a highly populated region. For this reason, numerous studies have been carried out over time to investigate the stress field and the geodynamic characteristics of the region using focal mechanisms. To provide a comprehensive set of revised information, which is challenging to build quickly because the data is dispersed over many papers, we collected and revised the focal mechanisms that were previously published in the

literature. Additionally, depending on the data quality and availability, we computed new focal mechanisms by first arrival polarity inversion or seismic moment tensor. Finally, we merged all the fault plane solutions to obtain a catalogue for a selection of 772 earthquakes with $1.8 \leq M \leq 6.5$. For each earthquake, we reported all the available focal mechanisms obtained by different authors. However, we also suggested a preferred solution for users who need expeditious information. The catalogue is available at https://doi.org/10.5281/zenodo.4284971 (Saraò et al., 2020).

**1 Introduction**

The focal mechanism describes the orientation of the fault on which an earthquake occurs and the direction of the slip. It is obtained by seismic waveforms using the polarities of the P waves first arrivals or retrieving the moment tensor. Currently, the preliminary focal mechanisms or fault plane solution (FPS) is calculated in near real time for seismic events with magnitudes (M) above 4 and published online by leading seismological agencies (e.g., the United States Geological Survey, European-

Mediterranean Seismological Centre, GEOFON Data Center). In the past, this was not the case, and the FPS was computed and published in studies investigating the geodynamics of specific regions. These studies are still being performed today to

---

[*] Andrea Restivo sadly passed away on 24 August 2020



revise the preliminary FPS or calculate the FPS of events of M < 4.0. The FPS is indeed necessary to investigate stress field to understand seismotectonic processes, but it is also essential for seismic hazard analysis.

The first methods to determine the source mechanisms of earthquakes (Knopoff and Gilbert, 1960) were based on observations

of the polarity of the first P wave motion at stations placed at known distances and azimuths from the source. Peak readings with alternating signs (compressions and dilatations) were plotted on the focal sphere (beach ball), and two orthogonal planes were drawn to separate the quadrants, representing the orientation of the fault and the auxiliary plane. The most commonly used graphical method employs a stereographic projection of the focal sphere, usually Schmidt or equal-area net projection. For many years, seismologists computed the FPS by graphical solutions manually based on the eye fit of the planes, but over

the years, the manual procedures have been replaced by computer-assisted programs (e.g., Whitcomb 1973; Buforn and Udías; 1984; Udías and Buforn 1988; Snoke et al., 1984; Reasenberger and Oppenheimer, 1985, Hardebeck and Shearer, 2002).

After the 1980s, with the development of digital broadband instruments, FPSs computed by moment tensors became very popular (e.g., Dziewonski et al., 1981). The first-order moment tensor is a mathematical representation of the equivalent body forces acting on a point source (Gilbert, 1970) and can be obtained by inversion either in the time or frequency domain of full

seismic waveforms or a selected part of them. The point source is located at the hypocentre, except for models that make use of the source centroid. The difference between the location of the initiation of rupture and the centroid can be significant, except for that of small earthquakes (Dziewonski and Woodhouse, 1983). For this reason, FPSs computed by polarities or by moment tensors might produce different results, not only due to systematic errors or inadequacies in the velocity models. The use of mechanisms by polarities represent the geometry of the fault at the initial breaking of the rupture, while the moment

tensor provides the source mechanism of the dominant component of the rupture geometry. Additionally, the differences between the two methods become relevant when the source deviates by the approximation of a pure double couple, for instance, when fluids play an essential role in earthquake generation. However, despite their limits - such as inadequacies in the P- and S-wave velocity models and poor station coverage, together with erroneous polarity readings that may result in large deviations between the model solution and the actual fault planes - the focal mechanisms using first motion polarities are still computed

and used. For small to moderate earthquakes (i.e., aftershock sequences), often they are the only source information available obtained using local network data (Shearer, 1998, Lentas et al., 2019).

At present, almost all seismological observatories compute quick moment tensors for earthquakes above approximately Mw 4.0 and publish solutions on dedicated online platforms. The Global Centroid-Moment-Tensor (CMT) Project (Dziewonski et al., 1981; Ekström et al., 2012), the National Earthquake Information Center (NEIC) of the USGS (Benz, 2017) and the

GEOFON data centre (2020) report moment tensor solutions for world seismicity, but there are also many regional or local moment tensor catalogues (e.g., Kubo et al., 2002; Pondrelli and Salimbeni 2015 (RCMT); Scognamiglio et al., 2009). Furthermore, some online databases, such as the bulletin of the International Seismological Centre (Lentas et al., 2019) or the database of the Stress World Map project (Zoback, 1992; Heidbach et al., 2018), contain both polarities and moment tensor FPSs of global seismicity.





Several authors have put considerable effort into researching FPSs reported in many papers and collecting them in catalogues for specific areas to provide a set of revised information, which is often challenging to build quickly. For European areas, in addition to the first compilations of Constantinescu et al. (1966), McKenzie (1972) and Udías et al. (1989), more recent catalogues include the EMMA database (Vannucci and Gasperini, 2004), which collects the focal mechanisms of earthquakes that occurred in the Mediterranean area from 1905 to 2006 where $4 \leq Mw \leq 8.7$.

In this study, we present a new catalogue of FPSs of earthquakes that occurred in the southeastern Alps and surrounding areas from 1928 to 2019 (Saraò et al., 2020). Because of the relevant seismic hazards (e.g., Slejko et al., 1998; Meletti et al., 2017) and its importance from a geodynamical point of view, many authors have computed the FPSs of earthquakes that occurred in this area using different data and different techniques (e.g., Anderson and Jackson, 1987; Slejko et al., 1989; Del Ben et al., 1991; Herak et al., 1995; Bernardis et al., 1997; Slejko et al., 1999; Aoudia et al., 2000; Pondrelli et al., 2001; Poli and Renner

2004; Danesi et al., 2015; Viganò et al., 2008; Bressan et al., 2007; 2009; 2018; Pondrelli and Salimbeni 2015; Restivo et al., 2016; Romano et al., 2019). A significant boost in seismological studies of the area occurred after the devastating 1976 Friuli earthquake (Ms=6.5), the strongest earthquake recorded in instrumental times in northeastern Italy (Slejko et al., 2018 and reference therein). In addition to the mainshocks, the most energetic aftershocks (e.g., Slejko et al., 1989; Slejko et al., 1999; Poli et al., 2002) were analysed to understand the geodynamic process occurring in the area; most of the FPSs of the Friuli

sequence were calculated manually using a graphical method. Subsequently, with the use of computer techniques (Whitcomb 1973; Reasenberger and Oppenheimer, 1985), many other small earthquakes belonging to the seismic sequences occurring in the area (e.g., Bernardis et al., 1997; Bressan et al., 2009; 2018) were carefully investigated to understand the stress regime of the area (e.g., Bressan et al., 2003; 2007; 2009; 2018). In recent times, Aoudia et al. (2000) and Pondrelli et al. (2011) reviewed the mechanisms of 1976 mainshocks in terms of the moment tensor, and since 2006, the moment tensor has been routinely

computed for earthquakes M≥3.6 in the region (e.g., Scognamiglio et al., 2009; Saraò 2016).

The aim of our study is to collect and revise all the FPSs published for the area over time in a comprehensive catalogue. Additionally, we employ a set of first polarity data collected by an author before data exchange via the Internet and visit select observatories to analyse the seismograms. These data were used to compute the FPS manually in previous studies, but only portions of them were published. We use the datasets to recalculate the FPS based on current knowledge (Sugan et al., 2020).

The new solutions are then merged with the FPS found in the literature. Depending on the data availability and quality, we compiled a dataset with the FPSs computed for 772 selected earthquakes (Saraò et al., 2020).

In the following section, we describe the study area, the methodologies and data used to build our catalogue and outline its main characteristics.

## 2 The study area

The region that we considered in this study (Fig. 1) is bounded by Garda Lake to the west, Western Slovenia to the east, the Venetian Po Plain to the south and Austria to the north (latitude 45°N-47.5°N and longitude 10°E-15°E). The area is one of



the most seismically active regions of the Alpine Belt and is involved in the convergence between the Adria microplate and Eurasia (e.g., Battaglia et al., 2004; D'Agostino et al., 2005, 2008; Serpelloni et al., 2005). The rates of convergence increase from west to the east by up to 1.5 mm/yr. to 2.0 mm/yr. (D'Agostino et al., 2005). The structural setting is a complex system

resulting from the superposition of several tectonic phases that have generated a NW-SE-trending Dinaric chain (to the east) and E-W-trending Alpine faults (southeastern Alps). On the west, the Giudicarie-Lessini region is a crucial zone for geodynamics, and it represents a tectonic boundary between the central-western and eastern Southern Alps with an orientation transverse to the strike of the Alpine chain (Viganò et al., 2015).

The National Institute of Oceanography and Applied Geophysics (OGS) northeastern Italy seismic and deformation network

(Priolo et al., 2005; Bragato et al., 2011, Bragato et al., 2020) has monitored the study region since 1977, complemented since 2002 by a GNSS network (Zuliani et al., 2018). Several studies (e.g., Gentili et al., 2011; Peruzza et al., 2015 and Sandron et al., 2018) provide a general description of the data recorded, including the main variations of the OGS North-Eastern Italy (OX) network geometry over time, the acquisition mode and the type of seismographs.

The seismicity, mainly located in the upper crust (Bressan et al., 2018, 2019; Viganò et al., 2015), is minor and directly related

to the deformation along the western margin of the Adriatic indentation (Gentili et al., 2011; Bressan et al., 2019; Romano et al., 2019). Rovida et al. (2020) reported that approximately thirty earthquakes of $5.5 \leq M \leq 6.5$ occurred in historical time, and the largest mainshocks (M>5.0) instrumentally recorded after the 1976 Ms=6.5 Friuli earthquakes (e.g., Aoudia et al., 2000) occurred in the 1998 Ms=5.7 and 2004 Mw=5.1 Bovec earthquakes (e.g., Bajc et al., 2001; Kastelic et al., 2008; Bressan et al., 2009) (Fig. 1).

## 110  3 Methodology

### 3.1 FPSs from the literature

The FPSs from the literature were obtained after careful research; the found FPSs were thoroughly checked for typos and orthogonality of the nodal planes as well as for the compatibility of pressure and tension axes with the nodal planes according to the Aki and Richards (1980) convention. The Aki and Richards (1980) convention defines the two planes by the strike,

measured clockwise from the north, with the fault dipping down to the right of the strike direction, and the dip, measured down from the horizontal. The rake is the angle between the strike direction and slip, where slip is taken as the direction of the hanging wall relative to the footwall. Because the two planes are orthogonal, the three angles defining one plane also define the orientation of the second plane. The orientations of the pressure (P) and tension (T) axes, located in the centre of the dilatational and compressional quadrants, respectively, are given by the azimuth (0-360°, North =0°, East=90°) and the plunge

(0-90°, down from the horizontal).

Starting with the published nodal planes, we recomputed the solutions to check their consistency and to add, when missing, a uniform dataset of parameters for each event. Whenever possible, we cross-checked the corrected solutions with the author or the beach ball shown in the original publication. We did not include the solutions for which it was impossible to recover





consistent planes starting from the available information in the catalogue. In the case of multiple FPSs for the same earthquake,
we considered all of them. However, when the FPS was copied from one paper to another by different authors, we reported
the FPS referred to in the first article it appeared. If the same authors published slightly different FPSs for the same earthquake,
we considered only the FPS of the most recent publication.

**3.2 The new computed FPSs**

Before computing the FPSs, we relocated the earthquakes using Hypo71 (Lee and Lahr, 1975), a standard location procedure,
and the 1D layered velocity model (Riggio and Russi, 1984; Bressan et al., 2003) used for the analysis of the earthquake
reported in the Friuli-Venezia Giulia Seismometric Network Bulletin (2020). The model had a P-wave velocity (Vp) = 5.85
km/s from the surface down to a depth of 22 km, a Vp = 6.8 km/s from 22 to 39.5 km depth and a Vp=8 km/s for the half-
space. The P to S-wave velocity ratio is 1.78.

To estimate the FPSs by polarities, we used the FPFIT algorithm (Reasenberger and Oppenheimer, 1985), which is based on
a grid-search procedure that finds the strike, dip and rake of the two planes by minimizing a normalized weighted sum of the
first-motion polarity discrepancies. The misfit function (zero perfect fit, one complete misfit) is computed from the number of
inconsistent polarities and weighted by the quality of the observation and the distance from the nodal planes. For some events,
the inversion program provided multiple solutions as a result of an insufficient number of polarity readings, the presence of
polarity misreading, the inclusion of localization errors or the use of an inadequate velocity model. In such cases, we selected
the preferred solution based on a) the data distribution on the focal sphere relative to the radiation pattern (i.e., number of first
motion polarities - FM ≥ 10; station distribution ratio - STDR ≥ 0.4; Misfit ≤ 0.3) and b) agreement with the tectonic lineaments
and style of the epicentral area.

The first polarities used to compute the FPSs were manually picked from seismograms or extracted from the *Bulletin of the
International Seismological Centre* and the *Seismological Bulletin of Slovenia*. The polarities were also read from the
seismograms archived in various Italian and European seismological observatories, many of which are no longer operating
(*Osservatorio meteorico-sismico nel Seminario - Chiavari; ENEL, Osservatorio Ximeniano - Florence, Osservatorio
Astronomico "Brera" - Milan, Dipartimento di Fisica dell'Università di Padova - Padua, Osservatorio S. Domenico – Prato,
Osservatorio meteoro-sismico nel Santuario di N.S. - Oropa, Osservatorio Bina - Perugia, Osservatorio "Valerio"- Pesaro,
Osservatorio meteorico-sismico nel Collegio Alberoni - Piacenza, Osservatorio Meteorico Istituto Fisica - University of
Siena, Sismografi Lungo Periodo di Mantovani (Bologna, Bolzano, Grosseto, Naples, Olbia, Palermo, Turin), Osservatorio
meteorico-sismico nel Seminario Maggiore - Treviso, Osservatorio meteorico-sismico nel Seminario Patriarcale – Venice,
Ljubljana, Munich, Stuttgart, Vienna).*

After 1977, the first motion polarities were also picked from the seismograms recorded by the network managed by OGS and
integrated with the data recorded by the surrounding seismic networks. Since 2006, we used the digital seismograms acquired
by the OX network to compute the seismic moment tensor for earthquakes $M_L$ ≥ 3.6 that occurred in NE Italy and the strict
surroundings (Bragato et al., 2011; Saraò et al., 2013; Saraò et al., 2016, Saraò, 2020). Using the TDMT code by Dreger



(2003), an algorithm widely employed in several observatories worldwide (e.g., Kubo et al., 2002; Clinton et al., 2006; Scognamiglio et al., 2009), we inverted the seismic waveforms in the frequency range of 0.02 - 0.05 Hz. A cross-correlation function was used to align data with Green's functions computed by the algorithm of Saikia (1984). This level of approximation

afforded a great level of flexibility in rapid source parameter determinations when the event locations and the origin time are preliminary (Dreger and Helmberger, 1993; Dreger et al., 1998). The source depth is estimated iteratively by finding the solution that yields the highest variance reduction (VR). VR is an index of the waveform fit between observed and synthetic seismograms and is given by the sum of squares of the difference in amplitude normalized by the observed waveforms (where 100% is best). Kubo et al. (2002) showed that stable and reliable solutions are obtained for VR greater than 50%.

Noisy or nodal stations and inadequate structural models can result in inaccurate moment tensor estimates. For this reason, Saraò (2007) performed a feasibility study to calibrate and tune the algorithm for the investigated area in relation to the station geometry and to the local velocity model employed (Bressan, 2005), finding the best parameter configuration that allows robust estimates of the best double couple orientation and of the $M_w$ value.

### 3.3 Merging the new FPSs with the published ones

Although we reported all the available FPSs for an earthquake in the catalogue, both retrieved from the literature or newly computed, we indicated a preferred one based on the following criteria: 1) the solution was computed within this study; 2) the solution was computed in the framework of a detailed study of the area and possibly after accurate relocation of the hypocentre. In both these cases, each fault plane solution was validated for data quality and distribution; 3) the fault plane solution was determined by moment tensor; 4) the solution is the latest computation; and 5) includes our knowledge of the main tectonic

features of the area. It was impossible to select the best solution based on the quality parameters of the FPS computation because such parameters were computed in a heterogeneous way and are provided sporadically.

To account for the variability in the solutions for the same event, we computed the 3D rotation angle by which one double couple was rotated into another arbitrary couple (Kagan, 1991). The Kagan angle varies between 0° for identical solutions and 120° for the absolute mismatches. Pairs of solutions with an angle below 20°-30° were considered very similar (Nakamura et

al., 2016; Lentas et al., 2019).

We used the software FMC (Álvarez-Gómez, 2019), which includes subroutines from Gasperini and Vannucci (2003), to verify and classify all the focal mechanisms in our catalogue. The fault type was classified by seven types of double couple classification (N: Normal; N-SS: Normal - Strike-slip; SS-N: Strike-slip - Normal; SS: Strike-slip; SS-R: Strike-slip - Reverse; R-SS: Reverse - Strike-slip; R: Reverse) similar to the conceptual geological classification of faults (Alvarez-Gomez, 2019).

To provide thorough information and facilitate easy inclusion in other databases, we also reported the classification adopted by the World Stress Map Project (Zoback 1992). This classified the events into five types (N, N-SS, SS, R-SS, R), and those not fitting in any of the five types were placed in the unknown category.





## 4 Analysis of the catalogue and discussion

We report in our catalogue 987 fault plane solutions for 772 earthquakes (Fig. 2) with a magnitude range of $1.8 \leq M \leq 6.5$

(Saraò et al., 2020). For each earthquake, we report the latitude and longitude of the epicentre, origin time, focal depth, magnitude, the strike, dip, rake of the two nodal planes, azimuth and plunge of the P, T, and B axes, fault type (using both Alvarez-Gomez, 2019 and Zoback 1992 classifications), Kagan angle and associated references. In the case of multiple solutions, we indicate the preferred one, and we indicate when the solution is computed by MT.

We collected and revised from the literature 836 FPSs (85% of the whole catalogue), 70 of which have been corrected with

respect to the original information, while for 341 we added the values missing to obtain a uniform dataset of information. Of the 151 newly computed focal mechanisms reported in the catalogue, 108 earthquakes with $1.8 \leq M \leq 4.8$ occurring between 1928 and 2019 are computed by first motion polarities (Table 1, Sugan et al. 2020 for details of the input data and inversion results), and 43 earthquakes with $3.4 \leq M_w \leq 5.1$ occurring from 2002 to 2018 are computed by moment tensor (Table 2, Saraò 2020, for details of the solutions).

The focal mechanisms are provided for selected earthquakes that occurred in the area based on data availability and quality, although since 1976, the MT is available by international agencies (e.g., CMT and RCMT) for all earthquakes $M \geq 4.5$, and since 2006, we compute the MT for all earthquakes of $M \geq 3.6$ and we publish the results online (RTS, 2020). For such reasons, the distribution in time and magnitude of the mechanisms of our catalogue may be uneven and linked to the study case of specific earthquakes and particular seismic sequences. Looking at the temporal distribution of the fault plane solutions of our

catalogue (Fig. 3), we observe peaks in seismicity following the 1976 Ms=6.5 Friuli earthquake, 1998 Ms=5.7 earthquake (Bajc et al., 2001) and 2004 Mw=5.1 2004 Bovec earthquakes (e.g., Bressan et al., 2007; 2009). The increasing number of focal mechanisms soon after the 1976 Friuli earthquake was boosted both by data coming from temporary stations (Slejko et al., 1999) and by the development of the OGS networks that has since then made it possible to investigate small energy earthquakes (M< 3.5). The magnitude of the computed FPSs ranges from 1.8 to 6.5, but most of the fault plane solutions have been

computed in the magnitude range of 2.8 to 3.5 (Fig. 4).

The ternary diagram of Fig. 5 shows the fault type distribution of the preferred focal mechanisms contained in our catalogue, while the pie plots (Fig. 6a, 6b) show the percentage of fault type mechanisms obtained by the two approaches to classify the mechanisms. The classification by Alvarez Gomez (Fig. 6a) accurately characterize seismic earthquakes with different strike-slip components. Reverse mechanisms feature the area under investigation, but the presence of strike-slip solutions is also

relevant and in agreement with the different kinematic regimes that characterize the region from east to west. Previous studies (e.g., Slejko et al., 1999; Poli et al., 2002; Bressan et al., 2018) have shown that thrust faulting is the dominant mechanism on the southeastern Alps showing a significant presence of strike-slip events, while strike-slip faulting prevails in the Dinaric domain (e.g., Bressan et al., 2018). Viganò et al. (2015) found that thrust faults with a strike-slip component and strike-slip faults prevail in the western sector, confirming that the seismotectonic zones Giudicarie and Lessini are undergoing different

kinematic regimes.



In Fig. 7, we plot the Kagan angle to show the difference between our preferred solutions and the other available FPSs. Although many pairs of focal mechanisms are below 20°-30°, there are discrepancies above 30°. The observed trend is not surprising and has been found for other datasets with multiple earthquake source solutions (e.g., Ishibe et al., 2014; Nakamura et al., 2016; Lentas et al., 2019) due to the differences in data and methods used to compute the solution for the same earthquake

over time.

From the comparison of the solutions computed in this study with solutions already published, we observe some agreement but also some discrepancies that are likely due to the different configuration parameters used to locate the events, the different datasets used as input and the diverse techniques employed.

For instance, the location of the 1928 Ms=5.8 Tolmezzo earthquake, the oldest event in our dataset (Table 1), is different from

that of previous studies (e.g., Slejko et al., 1989; Sandron et al., 2018), and it is more compatible both with the location reported in the macroseismic Italian database (Locati et al., 2016) and with the seismogenic features of the area than before (Bressan et al., 2018). However, the retrieved focal mechanism is a strike-slip type confirming the solutions previously found by other authors (e.g., Slejko et al., 1989; Cagnetti et al., 1976), probably because of the low impact of the hypocentral location when using data from very far-field stations, as was used in this case.

It is worth mentioning the results of the 1936 Ms=5.6 Alpago Cansiglio earthquake (Table 2), whose causative fault is still controversial (Galadini et al., 2005; Sugan and Peruzza 2011). We obtained two possible FPSs (Sugan et al., 2020): one with a strike-slip component, as previously found by others (e.g., Peruzza et al., 1989), and one with a compressive solution compatible within the uncertainties with the mechanism obtained by Sirovich and Pettenati (2004) using the regional macroseismic intensity pattern. We suggest the compressive solution as the preferred one and hypothesize that the new solution

can shed new light on the study of this earthquake being compatible with a known fault segment (Aviano Thrust outcropping) along the Cansiglio mountain front (Galadini et al., 2005).

## 5 Concluding remarks

We compiled a comprehensive catalogue of focal mechanisms for 772 selected earthquakes of $1.8 \leq M \leq 6.5$ that occurred in the southeastern Alps and surrounding areas from 1928 to 2019 (Saraò et al., 2020). The study region represents a key area

from a geodynamic point of view and is characterized by significant seismic risk, as reported in Italian seismic hazard maps. For such reasons, many authors have investigated the seismicity of this region, computing many focal mechanisms. In the catalogue, we have collected and revised 836 published solutions to provide a homogeneous and high quality dataset of focal mechanisms; 70 have been corrected for typos or inconsistencies, and, whenever possible, the corrected solutions have been discussed with the original author or checked against the available details in the paper. Additionally, we have enhanced and

made available the set of polarities readings of past earthquakes that would otherwise be lost; thus, we computed 108 new FPSs of earthquakes that occurred between 1928 and 2019 using a set of peak polarities readings that were not used or



published before for certain reasons (Sugan et al., 2020) and 43 earthquakes with $3.4 \leq Mw \leq 5.1$ that occurred from 2002 to 2018 by moment tensor (Saraò, 2020).

The distribution in time and magnitude of the FPSs are correlated to the study cases of specific earthquakes (e.g., 1976 Ms=6.5 Friuli earthquake, 1998 Ms=5.7 and 2004 Mw=5.1 Bovec earthquakes) and relevant seismic sequences occurred in the area. Thrust faulting is the dominant mechanism of our catalogue, with a significant presence of strike-slip events and a minor presence of normal faults. In the catalogue, we report all the FPSs available for each earthquake, and we suggest a preferred FPS for users who need to quickly have information to roughly represent the area at the current state of knowledge. By the Kagan angle, we quantify the difference among the preferred solutions and other solutions to provide additional insights to the final user. If the differences are well within the uncertainties for the majority of focal mechanism pairs, the discrepancy might be relevant for some other cases.

Our catalogue, available at https://doi.org/10.5281/zenodo.4284970, will be upgraded yearly with the FPSs of the most recent earthquakes occurring in the area to maintain the published dataset as up to date and complete as possible.

**Data**

The catalogue of focal mechanisms described in this paper is available at Zenodo (https://doi.org/10.5281/zenodo.4284971, Saraò et al., 2020).

The first motion dataset used to compute the focal mechanisms is available at Zenodo (https://doi.org/10.5281/zenodo.4284929, Sugan et al., 2020).

The seismic data used to compute the seismic moment tensor are provided by the OGS North-East Italy Seismic Network (OX) https://doi.org/10.7914/SN/OX. The waveforms can be downloaded from the European Integrated Data Archive EIDA (http://www.orfeus-eu.org/data/eida/).

**Author contributions**

AS designed the experiment. AS and MS wrote the manuscript, prepared the figures and revised the FPS collected from the literature. MS, GB, GF, AR read the first arrival peaks and computed the new FPS by polarities. AS computed the FPS by moment tensor. GB provided comments on the paper.

**Competing interests**

The authors declare that they have no conflict of interest.



**Acknowledgements**

We acknowledge the colleagues of the Seismological Research Centre of the OGS for the continuous effort in maintaining the
OGS North-Eastern Italy seismic and deformation network. We are grateful to Stefano Parolai for his insightful comments and
notable feedback.

**Financial support**

This research has been supported by the Regione Autonoma Friuli Venezia Giulia and by the Regione Veneto.

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

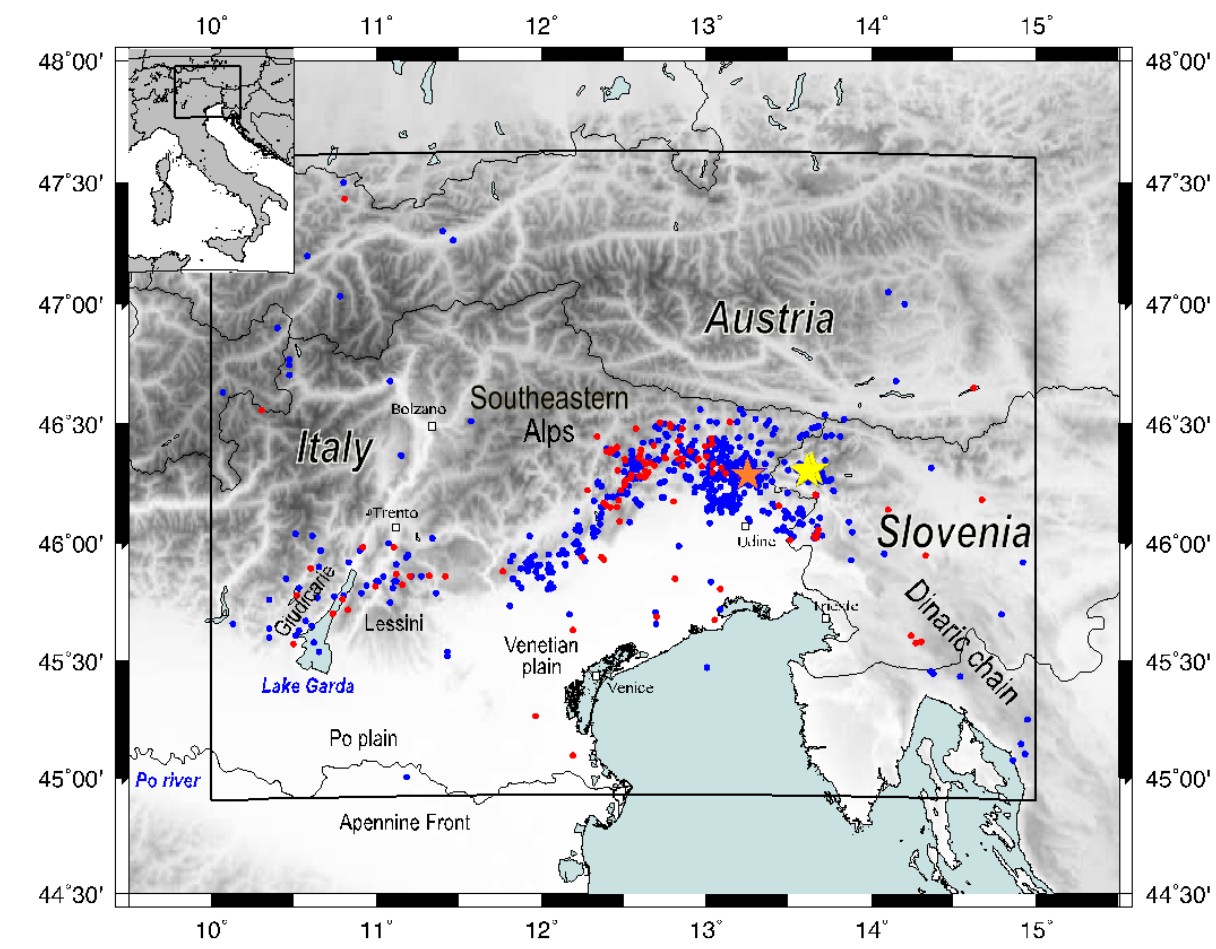

**Figure 1: Map of the epicentres of the focal mechanisms reported in the catalogue (Saraò et al., 2020) presented in this paper. The epicentres of the focal mechanisms retrieved from the literature are indicated by the blue dots, and the newly computed FPSs are indicated by the red dots. The 1976 Friuli earthquake (orange star) and the 1998 and 2004 Bovec earthquakes (yellow stars) are also shown. The map was generated by GMT software (Wessel et al., 2019).**




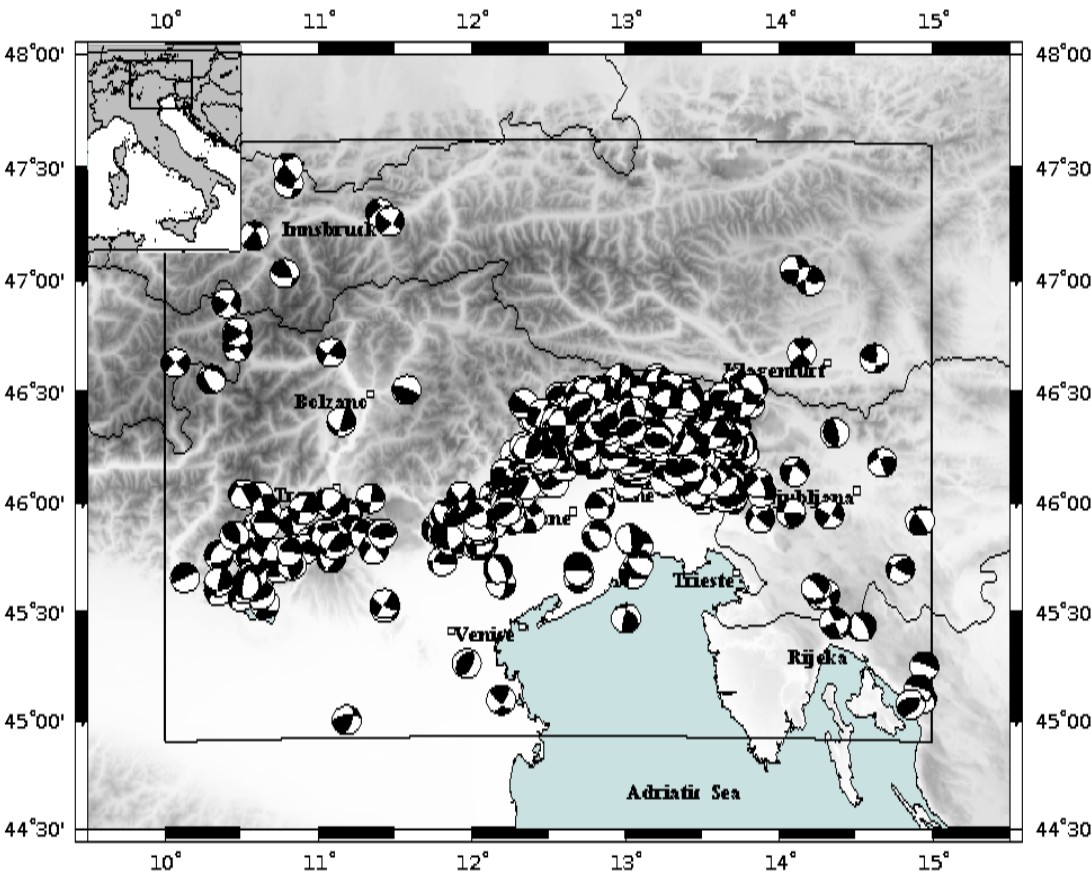

**Figure 2 - Overview of the focal mechanisms contained in our catalogue. The map was generated by GMT software (Wessel et al., 2019).**


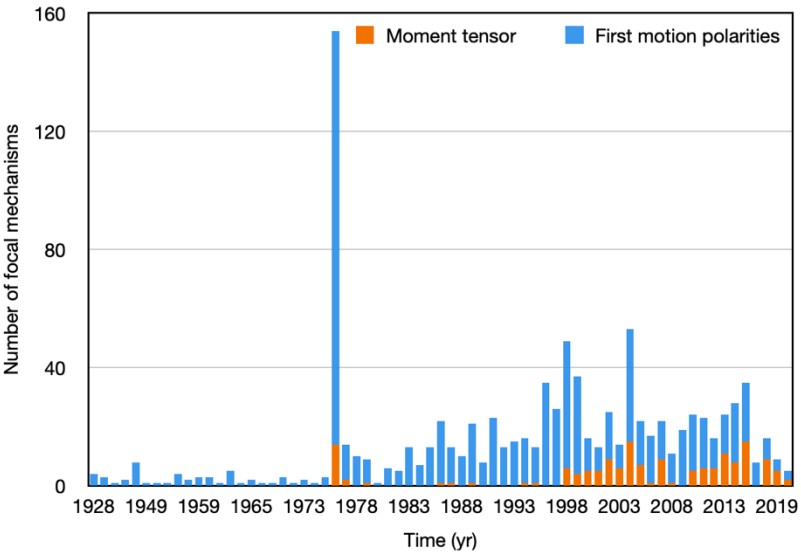

Figure 3 - Distribution in time of the fault plane solutions using a 1-year bin (year vs. number of fault plane solutions).

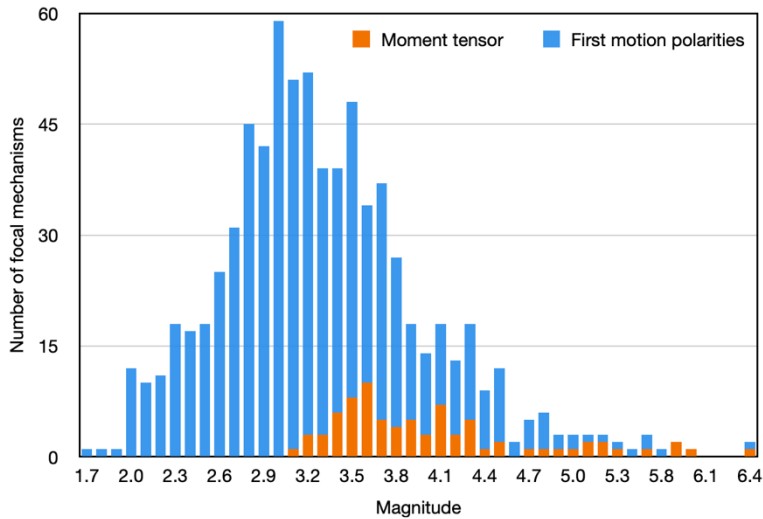

Figure 4 - Magnitude distribution of the fault plane solutions contained in the catalogue.

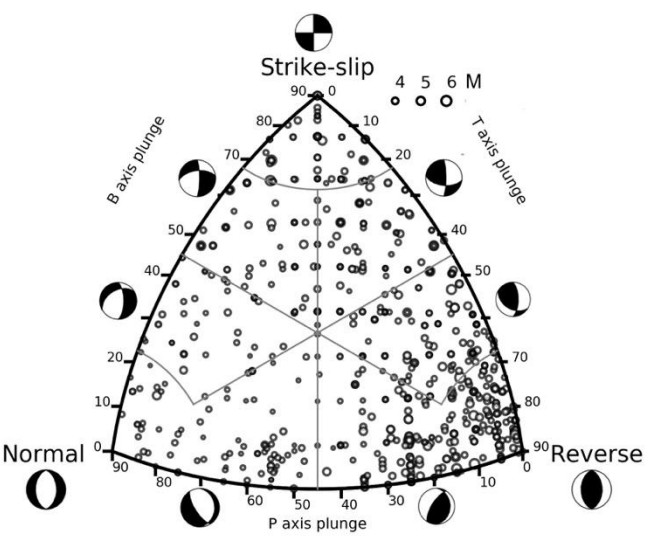


**Figure 5 –   Focal mechanism classification of our catalogue plotted on the ternary diagram (Kaverina et al., 1996).**

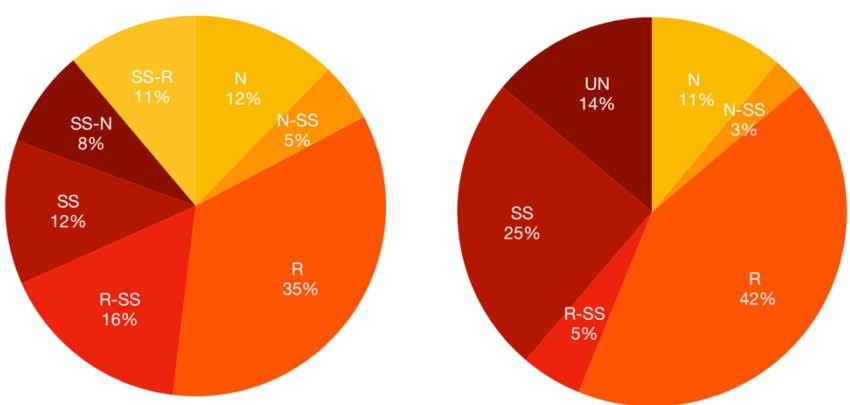

**Figure 6 - a) Classification of the focal mechanism according to Alvarez-Gomez (2019) and b) Zoback (1992). N = Normal; N-SS =**
**Normal - Strike-slip; SS-N= Strike-slip - Normal; SS = Strike-slip; SS-R= Strike-slip - Reverse; R-SS = Reverse - Strike-slip; R =**
**Reverse; UN = Unknown.**

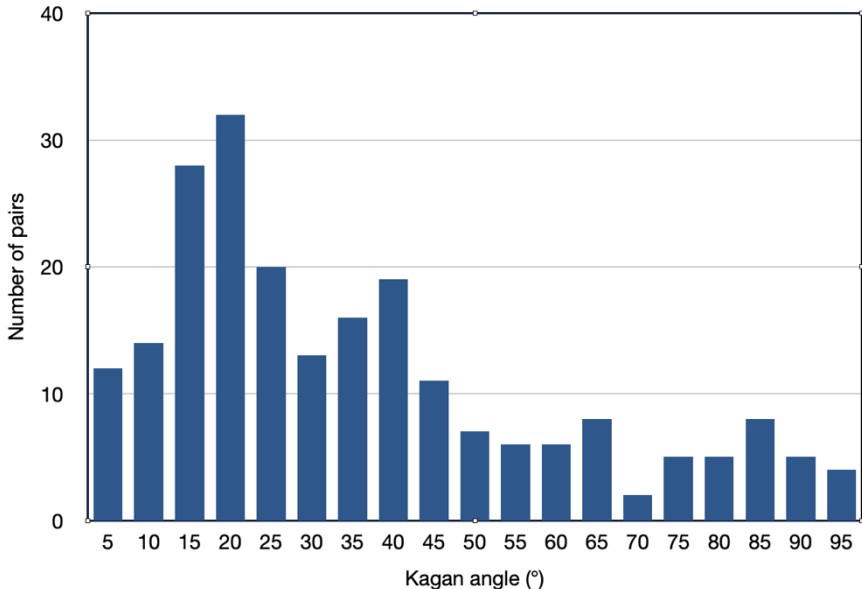

**Figure 7 - Kagan angle between the preferred solution and the multiple focal mechanism solutions reported in our catalogue.**




**Table 1 - Parameters of the new FPSs computed by first motion polarities (Sugan et al., 2020). Here we report: Date (yyyy-mm-dd), Time (hh:mm:ss), Lat. (latitude north in degrees) and Lon.( longitude east in degrees), Depth (km, * = fixed), M (magnitude), Str1, Dip1, Rak1 (strike, dip, rake of the first fault plane in degrees), Str2, Dip2, Rak2 (strike, dip, rake of the second fault plane in degrees), Ft (fault type according to Zoback 1992), FM (number of first motion polarities), STDR (station distribution ratio) and Misfit. Other details are given in the full catalogue (Saraò et al., 2020).**

| | Date | Time | Lat. | Lon. | Depth | M | Str1 | Dip1 | Rak1 | Str2 | Dip2 | Rak2 | Ft | Fm | STDR | Misfit |
|---|---|---|---|---|---|---|---|---|---|---|---|---|---|---|---|---|
| 1 | 1928-03-27 | 08:32:28 | 46.36 | 13.00 | 11.2 | 5.8 | 20 | 75 | -10 | 113 | 80 | -165 | SS | 23 | 0.7 | 0.1 |
| 2 | 1930-10-07 | 23:26:51 | 47.43 | 10.81 | 7* | 5.3 | 0 | 60 | 150 | 106 | 64 | 34 | TS | 20 | 0.7 | 0.0 |
| 3 | 1931-12-25 | 11:41:11 | 46.30 | 13.04 | 13.5 | 5.2 | 65 | 65 | 40 | 315 | 54 | 149 | TS | 11 | 0.6 | 0.0 |
| 4 | 1936-10-18 | 03:10:05 | 46.09 | 12.47 | 13.3 | 5.6 | 100 | 55 | 120 | 235 | 45 | 54 | TF | 37 | 0.5 | 0.1 |
| 5 | 1936-10-19 | 07:05:55 | 46.15 | 12.46 | 8.8 | 4.6 | 80 | 50 | 110 | 230 | 44 | 68 | TF | 13 | 0.4 | 0.0 |
| 6 | 1949-02-03 | 22:29:01 | 46.51 | 13.14 | 10* | 4.7 | 80 | 85 | -150 | 347 | 60 | -6 | SS | 16 | 0.8 | 0.1 |
| 7 | 1954-10-11 | 16:45:24 | 46.32 | 13.09 | 15.2 | 4.4 | 105 | 60 | 40 | 352 | 56 | 143 | TS | 11 | 0.7 | 0.0 |
| 8 | 1956-01-31 | 02:25:34 | 45.58 | 14.27 | 17.9 | 4.7 | 55 | 70 | 50 | 303 | 44 | 150 | TS | 13 | 0.6 | 0.0 |
| 9 | 1958-03-19 | 16:04:00 | 46.65 | 14.62 | 10.7 | 4.5 | 5 | 75 | 150 | 103 | 61 | 17 | SS | 13 | 0.6 | 0.0 |
| 10 | 1959-04-26 | 14:45:17 | 46.41 | 12.99 | 7.5 | 4.9 | 45 | 90 | 0 | 315 | 90 | 180 | SS | 33 | 0.5 | 0.1 |
| 11 | 1959-06-13 | 21:56:43 | 46.43 | 12.83 | 13.8 | 5 | 25 | 60 | -150 | 279 | 64 | -34 | NS | 16 | 0.4 | 0.1 |
| 12 | 1960-02-19 | 02:30:18 | 45.78 | 10.52 | 10.4 | 4.4 | 355 | 55 | 70 | 207 | 40 | 116 | TF | 20 | 0.5 | 0.1 |
| 13 | 1960-07-14 | 04:17:45 | 46.35 | 12.95 | 5.5 | 4.1 | 85 | 45 | 110 | 238 | 48 | 71 | TF | 13 | 0.4 | 0.0 |
| 14 | 1963-05-19 | 10:00:04 | 46.18 | 14.67 | 7.7 | 4.8 | 255 | 65 | 10 | 161 | 81 | 155 | SS | 33 | 0.5 | 0.1 |
| 15 | 1964-03-18 | 16:43:21 | 45.58 | 14.30 | 12.6 | 4.5 | 65 | 80 | 10 | 333 | 80 | 170 | SS | 23 | 0.7 | 0.1 |
| 16 | 1965-08-19 | 19:14:26 | 46.33 | 12.97 | 6.8 | 5 | 60 | 85 | 30 | 327 | 60 | 174 | SS | 16 | 0.8 | 0.0 |
| 17 | 1968-06-22 | 12:21:36 | 45.86 | 11.20 | 6.1 | 4.3 | 110 | 60 | 150 | 216 | 64 | 34 | TS | 30 | 0.7 | 0.1 |
| 18 | 1971-09-07 | 04:02:24 | 46.18 | 12.47 | 8.2 | 3.8 | 280 | 75 | -150 | 182 | 61 | -17 | SS | 14 | 0.6 | 0.0 |
| 19 | 1973-12-21 | 08:17:52 | 46.14 | 14.10 | 10.5 | 4 | 200 | 70 | -30 | 301 | 62 | -157 | SS | 22 | 0.4 | 0.2 |
| 20 | 1975-11-23 | 10:28:01 | 45.68 | 13.05 | 7* | 3.6 | 200 | 75 | -10 | 293 | 80 | -165 | SS | 17 | 0.6 | 0.1 |
| 21 | 1976-02-27 | 09:58:48 | 45.81 | 13.08 | 8.5 | 3.4 | 155 | 10 | -20 | 265 | 87 | -99 | U | 21 | 0.5 | 0.0 |
| 22 | 1979-11-06 | 03:04:00 | 46.22 | 12.28 | 9.6 | 3.8 | 75 | 80 | 170 | 167 | 80 | 10 | SS | 25 | 0.7 | 0.1 |
| 23 | 1981-04-15 | 20:35:08 | 46.32 | 12.85 | 10.5 | 3.5 | 85 | 55 | 140 | 201 | 58 | 42 | TS | 28 | 0.6 | 0.2 |
| 24 | 1981-06-17 | 18:55:31 | 46.38 | 12.82 | 7.7 | 3.1 | 80 | 55 | 130 | 204 | 51 | 47 | TF | 17 | 0.6 | 0.0 |
| 25 | 1981-06-28 | 08:42:54 | 46.48 | 12.85 | 7.5 | 3.5 | 105 | 65 | 120 | 231 | 38 | 43 | TF | 29 | 0.6 | 0.1 |
| 26 | 1981-12-05 | 05:47:40 | 46.34 | 12.65 | 7.5 | 4.5 | 35 | 25 | 20 | 287 | 82 | 114 | U | 43 | 0.6 | 0.1 |
| 27 | 1982-05-18 | 15:10:45 | 46.40 | 12.45 | 9.3 | 3.2 | 95 | 55 | -40 | 211 | 58 | -138 | NS | 25 | 0.6 | 0.1 |
| 28 | 1982-09-29 | 22:35:26 | 46.22 | 12.48 | 9.0* | 2.7 | 60 | 50 | 110 | 210 | 44 | 68 | TF | 26 | 0.5 | 0.1 |
| 29 | 1983-03-22 | 22:00:18 | 46.15 | 12.42 | 10.0 | 3 | 5 | 75 | 30 | 267 | 61 | 163 | SS | 27 | 0.6 | 0.2 |
| 30 | 1983-06-17 | 16:36:10 | 46.36 | 12.85 | 8.0 | 3.4 | 5 | 45 | 100 | 171 | 46 | 80 | TF | 35 | 0.6 | 0.2 |
| 31 | 1983-06-19 | 15:52:10 | 46.24 | 12.50 | 9.9 | 2.8 | 50 | 65 | 110 | 189 | 32 | 54 | TF | 25 | 0.5 | 0.1 |
| 32 | 1983-07-21 | 13:31:22 | 45.86 | 11.32 | 14.0 | 4.5 | 95 | 25 | 160 | 203 | 82 | 66 | U | 34 | 0.8 | 0.1 |
| 33 | 1984-07-08 | 07:58:49 | 45.63 | 12.19 | 17.2 | 3.6 | 5 | 65 | -170 | 271 | 81 | -25 | SS | 22 | 0.5 | 0.0 |
| 34 | 1984-10-25 | 13:58:55 | 45.61 | 14.24 | 12.2 | 3.5 | 0 | 10 | 90 | 180 | 80 | 90 | TF | 16 | 0.6 | 0.0 |
| 35 | 1984-10-29 | 13:29:26 | 46.26 | 12.51 | 10.9 | 3.3 | 105 | 35 | 80 | 297 | 56 | 97 | TF | 41 | 0.6 | 0.2 |



| | Date | Time | Lat. | Lon. | Depth | M | Str1 | Dip1 | Rak1 | Str2 | Dip2 | Rak2 | Ft | Fm | STDR | Misfit |
|---|---|---|---|---|---|---|---|---|---|---|---|---|---|---|---|---|
| 36 | 1984-12-15 | 10:55:10 | 46.28 | 12.60 | 10.0 | 3.7 | 70 | 60 | 110 | 214 | 36 | 59 | TF | 50 | 0.5 | 0.1 |
| 37 | 1985-02-08 | 01:45:52 | 46.50 | 12.78 | 6.1 | 3 | 115 | 80 | 150 | 211 | 61 | 12 | SS | 33 | 0.6 | 0.0 |
| 38 | 1985-02-08 | 21:10:40 | 46.49 | 12.78 | 7.0 | 2.7 | 25 | 80 | 20 | 291 | 70 | 169 | SS | 28 | 0.7 | 0.0 |
| 39 | 1985-05-05 | 17:55:31 | 46.34 | 12.62 | 11.3 | 2.9 | 235 | 75 | 140 | 337 | 52 | 19 | SS | 34 | 0.8 | 0.0 |
| 40 | 1985-06-18 | 04:52:56 | 45.82 | 11.00 | 12.5 | 3.6 | 95 | 60 | 140 | 208 | 56 | 37 | TS | 26 | 0.7 | 0.1 |
| 41 | 1985-07-09 | 23:09:48 | 46.51 | 12.72 | 7.0 | 2.4 | 20 | 70 | 50 | 268 | 44 | 150 | TS | 24 | 0.6 | 0.0 |
| 42 | 1985-08-04 | 06:59:12 | 46.48 | 12.57 | 8.0* | 2.3 | 75 | 70 | 90 | 255 | 20 | 90 | TF | 21 | 0.4 | 0.0 |
| 43 | 1985-11-24 | 05:36:17 | 46.35 | 12.50 | 5.8 | 2.3 | 60 | 50 | 80 | 255 | 41 | 102 | TF | 21 | 0.5 | 0.2 |
| 44 | 1985-11-24 | 06:28:34 | 46.35 | 12.51 | 6.7 | 2.7 | 90 | 40 | 120 | 233 | 56 | 67 | TF | 33 | 0.5 | 0.1 |
| 45 | 1986-01-13 | 12:50:38 | 46.36 | 12.74 | 8.7 | 2.1 | 60 | 45 | -130 | 290 | 57 | -57 | NF | 27 | 0.5 | 0.1 |
| 46 | 1986-01-15 | 01:40:17 | 46.16 | 12.39 | 6.6 | 2.8 | 200 | 30 | 40 | 74 | 71 | 114 | TF | 33 | 0.5 | 0.2 |
| 47 | 1986-02-05 | 22:52:50 | 46.30 | 12.65 | 3.9 | 3.1 | 80 | 55 | 10 | 344 | 82 | 145 | SS | 35 | 0.7 | 0.0 |
| 48 | 1986-07-05 | 10:33:18 | 46.38 | 12.39 | 7.0 | 2.4 | 105 | 45 | 120 | 246 | 52 | 63 | TF | 26 | 0.5 | 0.1 |
| 49 | 1986-09-04 | 20:50:49 | 46.40 | 12.44 | 7.0 | 2.6 | 130 | 30 | 150 | 247 | 76 | 63 | TF | 28 | 0.5 | 0.2 |
| 50 | 1986-10-07 | 20:59:59 | 46.39 | 12.42 | 6.2 | 2.4 | 135 | 30 | 90 | 315 | 60 | 90 | TF | 34 | 0.6 | 0.1 |
| 51 | 1986-10-08 | 20:18:45 | 46.38 | 12.41 | 4.3 | 2.9 | 125 | 35 | 80 | 317 | 56 | 97 | TF | 38 | 0.6 | 0.0 |
| 52 | 1987-03-10 | 23:16:26 | 46.39 | 12.60 | 7.2 | 2.5 | 90 | 50 | -120 | 312 | 48 | -59 | NF | 23 | 0.5 | 0.1 |
| 53 | 1987-04-07 | 20:04:20 | 46.45 | 12.34 | 7.0* | 3.6 | 60 | 40 | 20 | 314 | 77 | 128 | U | 52 | 0.6 | 0.2 |
| 54 | 1987-05-24 | 10:23:25 | 45.70 | 10.73 | 6.6 | 4.2 | 60 | 45 | 100 | 226 | 46 | 80 | TF | 46 | 0.5 | 0.2 |
| 55 | 1987-06-25 | 07:49:27 | 46.28 | 12.59 | 8.1 | 2.6 | 60 | 90 | -80 | 150 | 10 | -180 | U | 24 | 0.4 | 0.1 |
| 56 | 1987-07-10 | 08:09:28 | 45.98 | 10.92 | 8.7 | 3.7 | 85 | 55 | 140 | 201 | 58 | 42 | TS | 39 | 0.7 | 0.2 |
| 57 | 1987-10-20 | 00:33:26 | 46.30 | 12.63 | 3.7 | 3.4 | 275 | 90 | -20 | 5 | 70 | -180 | SS | 47 | 0.7 | 0.1 |
| 58 | 1987-10-31 | 13:09:41 | 46.35 | 12.83 | 8.6 | 2.8 | 20 | 45 | 120 | 161 | 52 | 63 | TF | 40 | 0.6 | 0.0 |
| 59 | 1987-12-04 | 14:45:12 | 45.89 | 10.60 | 7.1 | 3.8 | 125 | 80 | 120 | 232 | 31 | 19 | U | 35 | 0.6 | 0.1 |
| 60 | 1987-12-15 | 11:29:25 | 46.17 | 12.37 | 10.5 | 2.9 | 0 | 15 | 60 | 211 | 77 | 98 | TF | 32 | 0.6 | 0.1 |
| 61 | 1988-04-05 | 21:28:00 | 46.28 | 12.54 | 7.6 | 2.6 | 15 | 70 | -170 | 282 | 81 | -20 | SS | 29 | 0.6 | 0.1 |
| 62 | 1988-12-06 | 18:13:22 | 46.33 | 12.63 | 6.6 | 2.8 | 35 | 55 | 60 | 260 | 45 | 126 | TF | 27 | 0.5 | 0.2 |
| 63 | 1989-03-18 | 12:04:51 | 45.85 | 12.81 | 9.7 | 3.2 | 160 | 55 | -40 | 276 | 58 | -138 | NS | 28 | 0.6 | 0.0 |
| 64 | 1989-04-29 | 06:27:00 | 46.18 | 12.80 | 7.0* | 2.8 | 90 | 60 | -60 | 221 | 41 | -131 | NF | 18 | 0.5 | 0.1 |
| 65 | 1989-05-27 | 15:56:03 | 46.39 | 12.90 | 12.0 | 3.2 | 40 | 60 | 40 | 287 | 56 | 143 | TS | 13 | 0.5 | 0.2 |
| 66 | 1989-08-14 | 04:11:56 | 46.04 | 13.67 | 10.7 | 2.3 | 180 | 70 | -70 | 313 | 28 | -133 | NF | 18 | 0.5 | 0.0 |
| 67 | 1989-08-14 | 04:26:25 | 46.02 | 13.66 | 12.9 | 2.9 | 180 | 70 | -70 | 313 | 28 | -133 | NF | 22 | 0.6 | 0.1 |
| 68 | 1989-08-14 | 10:51:17 | 46.02 | 13.66 | 13.4 | 3 | 175 | 65 | -50 | 292 | 46 | -144 | NF | 22 | 0.6 | 0.1 |
| 69 | 1989-08-14 | 12:43:04 | 46.02 | 16.65 | 12.6 | 2.9 | 225 | 85 | -70 | 328 | 21 | -166 | U | 22 | 0.7 | 0.0 |
| 70 | 1990-02-04 | 08:13:13 | 46.21 | 13.66 | 13.2 | 2.4 | 180 | 45 | -50 | 310 | 57 | -123 | NF | 17 | 0.6 | 0.0 |
| 71 | 1990-02-04 | 09:22:15 | 46.20 | 13.66 | 13.3 | 2.6 | 185 | 45 | -40 | 306 | 63 | -127 | NF | 15 | 0.6 | 0.0 |
| 72 | 1990-06-28 | 18:56:59 | 45.94 | 12.36 | 11.3 | 2.5 | 240 | 85 | 160 | 332 | 70 | 5 | SS | 15 | 0.7 | 0.0 |
| 73 | 1990-06-28 | 19:30:10 | 45.93 | 12.38 | 11.2 | 3.2 | 60 | 90 | -170 | 330 | 80 | 0 | SS | 21 | 0.6 | 0.0 |
| 74 | 1992-02-24 | 21:31:43 | 46.30 | 12.46 | 9.7 | 2.6 | 60 | 70 | 120 | 181 | 36 | 36 | TF | 26 | 0.5 | 0.3 |
| 75 | 1992-03-11 | 15:40:32 | 45.95 | 14.33 | 10.3 | 3.9 | 30 | 80 | 0 | 300 | 90 | 170 | SS | 33 | 0.6 | 0.1 |
| 76 | 1992-07-13 | 09:40:58 | 46.06 | 13.68 | 12.3 | 2.3 | 280 | 85 | -130 | 184 | 40 | -8 | U | 16 | 0.5 | 0.1 |
| 77 | 1993-01-12 | 11:07:00 | 46.40 | 12.46 | 11.6 | 2.2 | 305 | 75 | -170 | 212 | 80 | -15 | SS | 16 | 0.7 | 0.1 |





| | Date | Time | Lat. | Lon. | Depth | M | Str1 | Dip1 | Rak1 | Str2 | Dip2 | Rak2 | Ft | Fm | STDR | Misfit |
|---|---|---|---|---|---|---|---|---|---|---|---|---|---|---|---|---|
| 78 | 1993-02-27 | 16:26:00 | 46.26 | 12.51 | 9* | 2.1 | 120 | 55 | 50 | 356 | 51 | 133 | TF | 13 | 0.7 | 0.0 |
| 79 | 1993-08-23 | 23:12:50 | 46.29 | 12.55 | 5.8 | 1.9 | 120 | 60 | -30 | 226 | 64 | -146 | NS | 16 | 0.7 | 0.2 |
| 80 | 1993-09-12 | 21:50:16 | 46.28 | 12.63 | 7.6 | 2 | 60 | 65 | 40 | 310 | 54 | 149 | TS | 19 | 0.6 | 0.2 |
| 81 | 1993-12-01 | 10:10:14 | 46.36 | 12.75 | 11.0 | 2.7 | 5 | 55 | 40 | 249 | 58 | 138 | TS | 16 | 0.7 | 0.1 |
| 82 | 1994-05-25 | 23:32:31 | 46.01 | 13.51 | 7.1 | 2.9 | 145 | 85 | -130 | 49 | 40 | -8 | U | 25 | 0.5 | 0.2 |
| 83 | 1994-12-05 | 21:14:09 | 46.41 | 12.68 | 10.8 | 2.6 | 35 | 80 | 20 | 301 | 70 | 169 | SS | 24 | 0.6 | 0.2 |
| 84 | 1996-02-10 | 04:02:55 | 45.82 | 11.15 | 14.8 | 3.1 | 65 | 50 | 80 | 260 | 41 | 102 | TF | 38 | 0.5 | 0.1 |
| 85 | 1997-03-17 | 22:45:12 | 46.44 | 13.03 | 7.6 | 3.1 | 90 | 20 | 60 | 302 | 73 | 100 | TF | 27 | 0.6 | 0.2 |
| 86 | 1997-06-16 | 14:38:26 | 45.88 | 11.99 | 12.6 | 2.8 | 55 | 25 | -60 | 203 | 69 | -103 | NF | 28 | 0.6 | 0.2 |
| 87 | 1997-06-19 | 16:54:09 | 45.94 | 12.25 | 9.6 | 2.5 | 115 | 75 | 130 | 222 | 42 | 23 | TS | 16 | 0.5 | 0.2 |
| 88 | 1997-07-06 | 22:28:58 | 45.57 | 10.50 | 11.6 | 3.5 | 220 | 70 | -50 | 332 | 44 | -150 | NS | 23 | 0.4 | 0.1 |
| 89 | 1997-07-25 | 15:54:18 | 45.87 | 11.12 | 13.7 | 3.3 | 135 | 50 | 130 | 262 | 54 | 53 | TF | 30 | 0.6 | 0.1 |
| 90 | 1997-10-18 | 19:58:45 | 45.10 | 12.19 | 24.5 | 3.2 | 130 | 70 | 20 | 33 | 71 | 159 | SS | 34 | 0.7 | 0.2 |
| 91 | 1998-12-26 | 19:30:58 | 45.86 | 11.42 | 15.3 | 3.6 | 90 | 60 | 100 | 251 | 31 | 73 | TF | 46 | 0.6 | 0.1 |
| 92 | 1999-01-05 | 03:22:15 | 45.76 | 10.79 | 5.0 | 3.3 | 35 | 85 | 40 | 301 | 50 | 173 | U | 39 | 0.4 | 0.1 |
| 93 | 1999-06-30 | 19:11:58 | 45.27 | 11.96 | 12.6 | 3.6 | 25 | 55 | 80 | 222 | 36 | 104 | TF | 33 | 0.5 | 0.1 |
| 94 | 2000-09-08 | 05:49:26 | 45.72 | 10.83 | 8.8 | 3.2 | 5 | 25 | 90 | 185 | 65 | 90 | TF | 25 | 0.5 | 0.2 |
| 95 | 2001-12-10 | 07:58:40 | 45.88 | 11.77 | 13.8 | 3.3 | 200 | 50 | -20 | 303 | 75 | -138 | NS | 29 | 0.6 | 0.2 |
| 96 | 2001-12-18 | 17:43:56 | 45.98 | 11.10 | 12.7 | 3.2 | 210 | 70 | -10 | 303 | 81 | -160 | SS | 36 | 0.6 | 0.2 |
| 97 | 2018-01-17 | 10:22:20 | 46.31 | 13.57 | 10.6 | 3.8 | 105 | 80 | 170 | 197 | 80 | 10 | SS | 35 | 0.7 | 0.1 |
| 98 | 2018-01-19 | 17:39:43 | 46.42 | 13.03 | 13.8 | 3.5 | 355 | 65 | 70 | 216 | 32 | 126 | TF | 36 | 0.6 | 0.1 |
| 99 | 2018-02-25 | 08:16:30 | 46.37 | 12.59 | 9.9 | 3.7 | 30 | 45 | 50 | 260 | 57 | 123 | TF | 35 | 0.6 | 0.1 |
| 100 | 2018-02-25 | 14:36:26 | 46.36 | 12.59 | 9.1 | 3.1 | 90 | 75 | -160 | 355 | 71 | -16 | SS | 32 | 0.7 | 0.1 |
| 101 | 2018-05-09 | 21:48:03 | 46.29 | 13.11 | 7.2 | 3.7 | 65 | 35 | 80 | 257 | 56 | 97 | TF | 38 | 0.5 | 0.1 |
| 102 | 2018-08-11 | 03:30:39 | 46.34 | 13.04 | 11.1 | 3.9 | 60 | 60 | 80 | 259 | 31 | 107 | TF | 40 | 0.5 | 0.1 |
| 103 | 2018-08-11 | 03:54:58 | 46.33 | 13.03 | 13.6 | 3.6 | 60 | 60 | 80 | 259 | 31 | 107 | TF | 38 | 0.5 | 0.1 |
| 104 | 2018-11-10 | 07:59:36 | 46.29 | 13.21 | 11.1 | 3 | 95 | 40 | 80 | 288 | 51 | 98 | TF | 33 | 0.5 | 0.1 |
| 105 | 2018-11-19 | 14:23:46 | 46.16 | 13.44 | 16.3 | 2.7 | 60 | 70 | 40 | 314 | 53 | 155 | TS | 34 | 0.6 | 0.1 |
| 106 | 2019-05-09 | 03:14:23 | 45.951 | 13.766 | 17.4 | 3.3 | 120 | 85 | -180 | 30 | 90 | -5 | SS | 36 | 0.6 | 0.2 |
| 107 | 2019-06-14 | 13:57:24 | 46.396 | 12.99 | 8.4 | 4 | 60 | 40 | 80 | 253 | 51 | 98 | TF | 32 | 0.5 | 0.1 |
| 108 | 2019-09-22 | 12:58:43 | 46.443 | 12.998 | 13.8 | 3.8 | 15 | 30 | 30 | 258 | 76 | 117 | TF | 31 | 0.6 | 0.1 |




**Table 2 - Parameters of the new focal mechanism solutions computed by the moment tensor (Saraò, 2020). Date (yyyy-mm-dd), Time (hh:mm:ss), Lat. (latitude north) and Lon. ( longitude east in degrees), Depth (km), Ml (local magnitude), MD (duration magnitude), Mw (moment magnitude), Str1, Dip1, Rak1 (strike, dip, rake of the first fault plane in degrees), Str2, Dip2, Rak2 (strike, dip, rake of the second fault plane in degrees), Ft (fault type according to Zoback 1992) and Q (quality inversion parameter; 4=best solution). Other details are given in the full catalogue (Saraò et al., 2020).**


| | Date | Time | Lat. | Lon. | Depth | Ml | $M_D$ | Mw | Mo (dyn cm) | Str1 | Dip1 | Rak1 | Str2 | Dip2 | Rak2 | Ft | Q |
|---|---|---|---|---|---|---|---|---|---|---|---|---|---|---|---|---|---|
| 1 | 2002-11-13 | 10:48:04 | 45.56 | 10.15 | 10 | | 4.2 | 4.1 | 1.36E+22 | 63 | 80 | 118 | 172 | 30 | 21 | U | 2 |
| 2 | 2004-07-12 | 13:04:00 | 46.30 | 13.63 | 4 | | 5.1 | 5.1 | 4.57E+23 | 220 | 83 | -9 | 311 | 81 | -173 | SS | 4 |
| 3 | 2005-01-14 | 07:58:13 | 46.19 | 14.02 | 12 | | 4.1 | 3.8 | 6.08E+21 | 201 | 68 | -29 | 303 | 63 | -156 | SS | 4 |
| 4 | 2005-01-14 | 08:05:19 | 46.20 | 14.04 | 14 | | 3.9 | 3.6 | 3.00E+21 | 216 | 90 | 1 | 126 | 89 | 180 | SS | 3 |
| 5 | 2005-04-24 | 18:34:01 | 45.56 | 14.29 | 8 | | 4.5 | 4.0 | 1.05E+22 | 155 | 77 | -165 | 62 | 75 | -13 | SS | 4 |
| 6 | 2007-01-01 | 14:59:45 | 46.51 | 14.23 | 10 | | 3.9 | 3.8 | 5.10E+21 | 85 | 50 | 91 | 263 | 40 | 89 | TF | 3 |
| 7 | 2007-02-05 | 08:30:04 | 45.11 | 15.00 | 10 | | 4.4 | 4.3 | 2.92E+22 | 224 | 72 | -26 | 322 | 66 | -161 | SS | 3 |
| 8 | 2007-05-02 | 12:49:13 | 46.50 | 14.47 | 8 | | 3.7 | 3.6 | 3.12E+21 | 80 | 55 | 81 | 275 | 36 | 102 | TF | 2 |
| 9 | 2007-05-19 | 16:19:40 | 47.17 | 10.61 | 14 | | 3.7 | 3.7 | 3.68E+21 | 338 | 89 | -5 | 68 | 85 | -179 | SS | 4 |
| 10 | 2007-08-13 | 13:58:30 | 45.18 | 13.45 | 12 | 3.6 | | 3.6 | 3.16E+21 | 168 | 84 | 140 | 263 | 51 | 8 | U | 2 |
| 11 | 2008-10-21 | 08:12:39 | 45.72 | 14.18 | 8 | 3.6 | | 3.4 | 1.50E+21 | 89 | 61 | 85 | 280 | 29 | 100 | TF | 1 |
| 12 | 2010-01-15 | 14:20:54 | 45.78 | 14.22 | 16 | 4.0 | | 3.5 | 1.91E+21 | 166 | 79 | -146 | 68 | 57 | -13 | SS | 3 |
| 13 | 2010-09-15 | 02:21:18 | 45.62 | 14.27 | 8 | 3.9 | | 3.6 | 3.33E+21 | 161 | 74 | 161 | 256 | 72 | 17 | SS | 4 |
| 14 | 2010-09-15 | 02:23:14 | 45.62 | 14.27 | 8 | 3.9 | | 3.5 | 1.96E+21 | 256 | 77 | 24 | 160 | 67 | 166 | SS | 4 |
| 15 | 2010-10-19 | 00:38:29 | 47.36 | 11.64 | 10 | 4.0 | | 3.5 | 1.91E+21 | 264 | 65 | 98 | 66 | 27 | 73 | TF | 2 |
| 16 | 2011-09-13 | 18:35:24 | 45.90 | 12.05 | 10 | 3.7 | | 3.4 | 1.50E+21 | 84 | 69 | 99 | 241 | 23 | 68 | TF | 3 |
| 17 | 2011-10-29 | 04:13:34 | 45.71 | 10.96 | 10 | 4.4 | | 4.0 | 9.74E+21 | 245 | 51 | 79 | 81 | 40 | 103 | TF | 3 |
| 18 | 2012-01-24 | 23:54:46 | 45.55 | 11.00 | 10 | 4.2 | | 4.0 | 1.18E+22 | 199 | 86 | 29 | 107 | 61 | 176 | SS | 3 |
| 19 | 2012-05-29 | 18:28:04 | 45.06 | 11.05 | 6 | 3.8 | | 4.0 | 1.17E+22 | 265 | 68 | 83 | 102 | 23 | 106 | TF | 4 |
| 20 | 2012-06-09 | 02:04:56 | 46.20 | 12.47 | 6 | 4.3 | | 4.1 | 1.86E+22 | 54 | 69 | 92 | 227 | 21 | 84 | TF | 4 |
| 21 | 2012-12-03 | 04:36:00 | 46.23 | 14.81 | 18 | 4.2 | | 3.9 | 8.22E+21 | 47 | 78 | 24 | 311 | 66 | 167 | SS | 3 |
| 22 | 2013-02-02 | 13:35:33 | 46.48 | 14.63 | 4 | 4.3 | | 4.2 | 2.06E+22 | 96 | 60 | 58 | 328 | 43 | 133 | TF | 3 |
| 23 | 2013-02-12 | 18:12:25 | 46.28 | 12.58 | 18 | 3.8 | | 3.7 | 3.66E+21 | 234 | 51 | 64 | 91 | 45 | 118 | TF | 3 |
| 24 | 2013-06-16 | 20:04:58 | 45.77 | 14.84 | 2 | 4.0 | | 3.7 | 4.30E+21 | 253 | 53 | 59 | 117 | 47 | 124 | TF | 4 |
| 25 | 2013-07-30 | 12:58:28 | 45.00 | 15.10 | 16 | 4.7 | | 4.3 | 2.71E+22 | 212 | 86 | 17 | 121 | 73 | 176 | SS | 4 |
| 26 | 2013-08-24 | 13:59:01 | 46.21 | 12.55 | 10 | 3.6 | | 3.5 | 2.04E+21 | 55 | 71 | 93 | 225 | 20 | 81 | TF | 3 |
| 27 | 2014-04-22 | 08:58:27 | 45.65 | 14.24 | 12 | 4.7 | | 4.4 | 4.43E+22 | 249 | 87 | -7 | 340 | 83 | -177 | SS | 4 |
| 28 | 2014-05-29 | 07:24:18 | 46.10 | 13.86 | 12 | 3.8 | | 3.6 | 2.79E+21 | 225 | 72 | -22 | 322 | 69 | -160 | SS | 3 |
| 29 | 2015-01-30 | 00:45:49 | 46.39 | 13.15 | 12 | 4.1 | | 3.9 | 8.95E+21 | 53 | 81 | 72 | 297 | 20 | 153 | U | 4 |
| 30 | 2015-05-12 | 02:02:50 | 45.89 | 12.05 | 8 | 3.5 | | 3.5 | 2.16E+21 | 59 | 69 | 85 | 254 | 21 | 104 | TF | 3 |
| 31 | 2015-05-15 | 05:35:47 | 45.88 | 12.06 | 6 | 3.6 | | 3.5 | 2.23E+21 | 76 | 74 | 96 | 235 | 17 | 70 | TF | 3 |
| 32 | 2015-08-01 | 20:47:52 | 45.90 | 10.78 | 2 | 3.8 | | 3.5 | 2.08E+21 | 355 | 77 | 32 | 257 | 59 | 165 | SS | 3 |
| 33 | 2015-08-18 | 20:10:02 | 45.89 | 11.90 | 12 | 3.6 | | 3.4 | 1.57E+21 | 70 | 83 | 82 | 298 | 11 | 137 | U | 3 |
| 34 | 2015-08-29 | 18:47:04 | 46.31 | 13.60 | 6 | 4.3 | | 4.1 | 1.56E+22 | 82 | 65 | 66 | 310 | 34 | 132 | TF | 3 |
| 35 | 2015-11-01 | 07:52:32 | 45.83 | 15.64 | 6 | 4.8 | | 4.3 | 3.10E+22 | 273 | 61 | 83 | 106 | 30 | 102 | TF | 4 |



| | Date | Time | Lat. | Lon. | Depth | Ml | M$_D$ | Mw | Mo (dyn cm) | Str1 | Dip1 | Rak1 | Str2 | Dip2 | Rak2 | Ft | Q |
|---|---|---|---|---|---|---|---|---|---|---|---|---|---|---|---|---|---|
| 36 | 2015-11-21 | 11:52:38 | 46.43 | 12.71 | 6 | 3.5 | | 3.6 | 2.49E+21 | 243 | 51 | 100 | 47 | 40 | 78 | TF | 2 |
| 37 | 2017-06-04 | 18:00:57 | 45.65 | 10.71 | 6 | 3.7 | | 3.4 | 1.31E+21 | 245 | 51 | 112 | 33 | 44 | 66 | TF | 2 |
| 38 | 2017-07-21 | 17:03:56 | 45.65 | 10.70 | 6 | 3.4 | | 3.4 | 1.44E+21 | 220 | 77 | 66 | 102 | 27 | 150 | TF | 1 |
| 39 | 2017-09-06 | 12:22:30 | 46.27 | 11.99 | 6 | 3.6 | | 3.6 | 2.45E+21 | 300 | 78 | 148 | 38 | 59 | 14 | SS | 3 |
| 40 | 2018-01-17 | 10:22:20 | 46.32 | 13.58 | 8 | 3.8 | | 3.8 | 5.31E+21 | 32 | 77 | 15 | 299 | 75 | 167 | SS | 3 |
| 41 | 2018-02-25 | 08:16:29 | 46.37 | 12.60 | 10 | 3.7 | | 3.7 | 3.56E+21 | 8 | 88 | -16 | 98 | 74 | -178 | SS | 3 |
| 42 | 2018-08-11 | 03:30:39 | 46.34 | 13.07 | 8 | 3.9 | | 3.7 | 3.88E+21 | 52 | 75 | 55 | 301 | 38 | 154 | U | 3 |
| 43 | 2018-12-05 | 16:23:59 | 45.69 | 14.28 | 8 | 3.8 | | 3.5 | 2.11E+21 | 30 | 75 | -62 | 146 | 31 | -150 | NS | 1 |