# Peer review of "A focal mechanism catalogue of earthquakes that occurred in the southeastern Alps and surrounding areas from 1928 – 2019"

_Earth System Science Data, 2020_

## Referee Comment (RC1)

**Review**

**Title:** A focal mechanism catalogue of earthquakes that occurred in the southeastern Alps and surrounding areas from 1928 – 2019

**Authors**: Angela Saraò, Monica Sugan, Gianni Bressan, Gianfranco Renner, Andrea Restivo

**Manuscript**: ESSD2020-369, https://doi.org/10.5194/essd-2020-369

**General remarks and comments:**

The article is appropriate and the dataset is meaningful and useful. The dataset can be used in the current format and dimensions. Check only a few FMs to avoid inconsistencies. Figures and tables are correct and of good quality.
I really appreciate this kind of works that are fundamental tools for many analyses, as mentioned by the Authors. Data catalogs are not often seen as a novelty by the scientific community but hide an big effort for the rigorous data preparation and deserve much more visibility and resources.
I have only minor revisions to suggest and a check of parameters of few FMs.
More comments are in the following review.

**Abstract**
Line 18: delete: "However, …"

**Introduction**:
I suggest to underline the great effort to prepare catalogues like the proposed one: collect, select parameters, standardize the FMs information and also elaborate new FPS solutions. This can be a useful reminder for FMs databases users. I also suggest highlighting and citing others datasets of FMs that assess a preferred solution too (e.g. Custodio et al., 2016; Kapetanidis and Kassaras, 2019 and not only Vannucci and Gasperini, 2004).

Line 53-54: "*At present, almost all seismological observatories compute quick moment tensors for earthquakes above approximately Mw 4.0..*" specify the thresholds of magnitudes for catalogues (e.g. Mw 5.0-5.5 for GCMT and so on). Also at line 24, the USGS has a higher threshold of Magnitude. Maybe you should cite the TDMT Catalogue of INGV.

Line 56: "*…local moment tensor catalogues..*": add acronyms and information of the catalogues (RCMT, Regional Centroid Moment Tensor Catalog. Scognamiglio et al., 2009 provide the TDMT, Time Doman Moment Tensor catalogue. Moreover change the reference Scognamiglio et al., 2009 with Scognamiglio et al., 2006 (http://terremoti.ingv.it/tdmt)

Lines 58-59: "*database of the Stress World Map project (Zoback, 1992; Heidbach et al., 2018), contain both polarities and moment tensor FPSs of global seismicity.*". This is correct as a general introduction, but in the database no one focal mechanisms is taken from these authors.

Line 78: "*Pondrelli et al. (2011)*" maybe 2001?
Lines 174-175: "*includes our knowledge of the main tectonic features of the area*": I think the sentence is a bit vague, too subjective and not very transparent. Moreover, the Authors could calculate a weight in the database (e.g. criterion 1=100000, criterion 2=10000, criterion 3=1000 and so on). This explains in detail the choice of the preferred solution. Alternatively you can classify the preferred solution: P1, P2, P3, P4, P5.

Lines 177-180: Add a brief description on the "subjects" of the rotation, i.e. the preferred solution and the alternative ones (i.e. solutions of other authors) for the same earthquake. Caption of Fig 7: "*...and the multiple focal mechanism solutions*": multiple is not appropriate, change to "alternative"

Lines 194-195: "*…70 of which have been corrected with respect to the original information…*". Add a comment in the database about the changes.
Line 201: "*CMT*" is GCMT ?

Lines: 216-218: avoiding the second reference of Bressan et al, 2018 in the same sentence. Add the reference Serpelloni et al., 20116 (Tectonophysics) which also investigates the tectonic regime of this area.

Line 224: "*…other available FPSs*", change in: ….other FPSs available for the same earthquake.

Line 231: Locati et al., 2016 can be change to CPTI15v3. Note that the magnitude of CPTI15 is Mw 6.08

**Concluding remarks**
Lines 243-252: specify that the database collects 936 focal solutions.

**References:**
Review the references and the correspondence with the text (e.g. Serpelloni et al., 2005 https://doi.org/10.1111/j.1365-246X.2005.02618.x is double)

I suggest moving the reference to GMT software from the text to the Acknowledgements

**Database:** "Focal mechanisms of the southeastern Alps and surroundings" available at https://doi.org/10.5281/zenodo.4284971

Authors could verify the parameters and the correspondence among the parameters to avoid inconsistencies. For example some FMs of Sugan et al. (2020) have negative B axis plunge (instead of the range: 0-90). In a few cases the nodal planes and the axes are not in agreement each other and the differences of parameters (e.g. some parameters of the nodal plane B derived from the nodal plane A are > 4 degrees, as well as some P and T axes derived from nodal planes or viceversa) (see table below)

| Source | day | month | year |
|---|---|---|---|
| Muller, 1977 | 06 | 05 | 1976 |
| Eva & Pastore, 1993 | 13 | 09 | 1989 |
| Poli et al., 2002 | 05 | 10 | 1991 |
| Bressan et al., 2018 | 01 | 06 | 2009 |
| Restivo et al., 2016 | 28 | 10 | 2010 |
| ISC | 02 | 02 | 2013 |
| Romano et al., 2019 | 02 | 05 | 2013 |
| Romano et al., 2019 | 12 | 05 | 2015 |
| Bressan et al., 2018 | 23 | 11 | 2017 |

I suggest to eliminate trend and plunge of B axis from the catalogue and to investigate about some solutions (table below) to improve the database:

Gianfranco Vannucci, INGV

---

## Author Response (AR1)

**Reply to the reviewer's comments on the paper "A focal mechanism catalogue of earthquakes that occurred in the southeastern Alps and surrounding areas from 1928 – 2019", Manuscript: ESSD2020-369, https://doi.org/10.5194/essd-2020-369**

Authors: Angela Saraò, Monica Sugan, Gianni Bressan, Gianfranco Renner, Andrea Restivo

We would first like to express our gratitude for the valuable time that both the reviewers devoted to our paper and for the helpful suggestions that will certainly improve our work.
We have addressed all the issues raised, both in the manuscript and in the database.

In the following, the detailed replies to the reviewer's comments

**Rev. 1**

**General remarks and comments:**
The article is appropriate and the dataset is meaningful and useful. The dataset can be used in the current format and dimensions. Check only a few FMs to avoid inconsistencies. Figures and tables are correct and of good quality.
I really appreciate this kind of works that are fundamental tools for many analyses, as mentioned by the Authors. Data catalogs are not often seen as a novelty by the scientific community but hide an big effort for the rigorous data preparation and deserve much more visibility and resources. I have only minor revisions to suggest and a check of parameters of few FMs. More comments are in the following review.

Thanks for the recognition.

**1) Abstract**
Line 18: delete: "However, …"

Corrected.

**2) Introduction:**
I suggest to underline the great effort to prepare catalogues like the proposed one: collect, select parameters, standardize the FMs information and also elaborate new FPS solutions. This can be a useful reminder for FMs databases users. I also suggest highlighting and citing others datasets of FMs that assess a preferred solution too (e.g. Custodio et al., 2016; Kapetanidis and Kassaras, 2019 and not only Vannucci and Gasperini, 2004).

In the Introduction we added the following sentences:

*"Several authors have put considerable effort into researching FPS reported in many papers and collecting them in catalogues for specific areas to provide a set of revised information, which is often challenging to build quickly. Valuable to stress that collecting data, very often spread in different documents and locations, checking and selecting parameters, standardize the information is a long and painstaking job, sometimes not fully known.*

*For European areas, in addition to the first compilations of Constantinescu et al. (1966), McKenzie (1972) and Udías et al. (1989), more recent catalogues include the EMMA database (Vannucci and Gasperini, 2004), which collects the focal mechanisms of earthquakes that occurred in the Mediterranean area from 1905 to 2006 in the range $4 \leq Mw \leq 8.7$. In several cases, after merging the available FPS for an earthquake, the authors assess and suggest a preferred solution based on different priorities or strategy (e.g. Gerner, 1995; Radulian et al., 2002; Custódio et al., 2016; Kapetanidis and Kassaras, 2019)."*

**3) Line 53-54: "At present, almost all seismological observatories compute quick moment tensors for earthquakes above approximately Mw 4.0.." specify the thresholds of magnitudes for catalogues (e.g. Mw 5.0-5.5 for GCMT and so on). Also at line 24, the USGS has a higher threshold of Magnitude. Maybe you should cite the TDMT Catalogue of INGV.**

We changed as following:

*"At present, almost all seismological observatories compute quick moment tensors for earthquakes above a certain threshold of magnitude and publish solutions on dedicated online platforms. The Global Centroid-Moment-Tensor (CMT) Project (Dziewonski et al., 1981; Ekström et al., 2012), the National Earthquake Information Center (NEIC) of the USGS (Benz, 2017) and the GEOFON data centre (2020) report moment tensor solutions for world seismicity and thresholds of magnitudes of about Mw 5.0, Mw 5.8 and Mw 4.5 respectively."*

4) Line 56: "…local moment tensor catalogues..": add acronyms and information of the catalogues
(RCMT, Regional Centroid Moment Tensor Catalog. Scognamiglio et al., 2009 provide the TDMT,
Time Doman Moment Tensor catalogue. Moreover change the reference Scognamiglio et al., 2009 with Scognamiglio et al., 2006 (http://terremoti.ingv.it/tdmt)"

We add acronyms and information of the catalogues as you suggested:
*"In addition to these, there are also many regional or local moment tensor catalogues with magnitude thresholds of Mw 3.6 (e.g Scognamiglio et al., 2006 [Time Domain Moment Tensor catalogue – TDMT]; Kubo et al., 2002 [NIED seismic moment tensor catalogue) and Mw 4.5. (e.g. Pondrelli and Salimbeni 2015 [Regional Centroid Moment Tensor Catalog – RCMT]."*
The reference Scognamiglio et al., 2009, has been changed with the reference Scognamiglio et al., 2006:
Scognamiglio, L., Tinti, E., Quintiliani, M.: Time Domain Moment Tensor (TDMT) [Data set], Istituto Nazionale di Geofisica e Vulcanologia (INGV), https://doi.org/10.13127/TDMT, 2006.

5) Lines 58-59: "database of the Stress World Map project (Zoback, 1992; Heidbach et al., 2018),
contain both polarities and moment tensor FPSs of global seismicity." This is correct as a general introduction, but in the database no one focal mechanisms is taken from these authors.

You are right. Actually, we would like to give general information of a very global catalogue used for a specific purpose.

6) Line 78: "Pondrelli et al. (2011)" maybe 2001?

Yes, it has been corrected.

7 ) Lines 174-175: "includes our knowledge of the main tectonic features of the area": I think the sentence is a bit vague, too subjective and not very transparent. Moreover, the Authors could calculate a weight in the database (e.g. criterion 1=100000, criterion 2=10000, criterion 3=1000 and so on). This explains in detail the choice of the preferred solution. Alternatively you can classify the preferred solution: P1, P2, P3, P4, P5.

You are right. We added in the database a Priority criteria code for each solution. The preferred one has now the P followed by the priority code as described in the text. We have modified the text in the paper accordingly.
*"Although we reported all the available FPSs for an earthquake in the catalogue, both retrieved from the literature or newly computed, we indicated a preferred one based on the following priority criteria: 1) the solution was computed within this study; 2) the fault plane solution was determined by moment tensor; 3) the solution was computed in the framework of a detailed study of the area and possibly after accurate relocation of the hypocenter. In both these cases, each fault plane solution was validated for data quality and distribution; 4) the solution is the latest computation; and 5) the solution is compatible with includes our knowledge of the main tectonic features of the area: 6) the solution is part of a regional catalogue; 7) the solution is compatible with most of the solutions proposed by independent studies (i.e. Kagan angle)."*
….

**4 Analysis of the catalogue and discussion**
….
*In the case of multiple solutions, we indicate the preferred one, and the classification according the priority code described in the previous section.*
…

8) Lines 177-180: Add a brief description on the "subjects" of the rotation, i.e. the preferred solution and the alternative ones (i.e. solutions of other authors) for the same earthquake.

We have clarified in the manuscript as in the following:
*"To account for the variability between the preferred solution and the alternative ones for the same event, we computed the 3D rotation angle by which one double couple was rotated into another arbitrary couple (Kagan, 1991)."*

9) Caption of Fig 7:
"…and the multiple focal mechanism solutions": multiple is not appropriate, change to "alternative"

Done.

10) Lines 194-195: "…70 of which have been corrected with respect to the original information…".
Add a comment in the database about the changes.

Thanks for bringing this to our attention. We added a new column in the database with comments. For each modified solution we added in the comments the done change.

11) Line 201: "CMT" is GCMT ?

The Global Centroid-Moment-Tensor is referred as "CMT" at https://www.globalcmt.org/.

12) Lines: 216-218: avoiding the second reference of Bressan et al, 2018 in the same sentence. Add the reference Serpelloni et al., 20116 (Tectonophysics) which also investigates the tectonic regime of this area.

Done.

13) Line 224: "…other available FPSs", change in: ….other FPSs available for the same earthquake.

Done.

14) Line 231: Locati et al., 2016 can be change to CPTI15v3. Note that the magnitude of CPTI15 is Mw 6.08

We kept Locati et al. 2016 in the text, since we address to Italian Macroseismic database, but added text and reference about the CPTI15v3, as you suggested.

*"For instance, the location of the 1928 Ms=5.8 Tolmezzo earthquake, the oldest event in our dataset (Table 1), is different from that of previous studies (e.g., Slejko et al., 1989; Sandron et al., 2018), and it is more compatible both with the location reported in the macroseismic Italian database and associated catalogue (Locati et al., 2016, Rovida et al., 2020; 2020) and with the seismogenic features of the area than before (Bressan et al., 2018). The revised magnitude for this event to be equal to 6.08, in Rovida et al., (2021)."*
Reference added:
Rovida, A., Locati, M., Camassi, R., Lolli, B., Gasperini, P., Antonucci, A.: Catalogo Parametrico dei Terremoti Italiani (CPTI15), versione 3.0. Istituto Nazionale di Geofisica e Vulcanologia (INGV), https://doi.org/10.13127/CPTI/CPTI15.3, 2021.

15) Concluding remarks
Lines 243-252: specify that the database collects 936 focal solutions.

Done.

16) **References**:
Review the references and the correspondence with the text (e.g. Serpelloni et al., 2005 https://doi.org/10.1111/j.1365-246X.2005.02618.x is double)

The reference of Serpelloni has been corrected.

17) I suggest moving the reference to GMT software from the text to the Acknowledgements

Done. However, we need to keep the reference also in the captions of Fig. 1 and Fig. 2 since we were explicitly requested by the editorial office to specify the origin of the maps.

18) **Database**
Authors could verify the parameters and the correspondence among the parameters to avoid inconsistencies. For example some FMs of Sugan et al. (2020) have negative B axis plunge (instead of the range: 0-90). In a few cases the nodal planes and the axes are not in agreement each other and the differences of parameters (e.g. some parameters of the nodal plane B derived from the nodal plane A are > 4 degrees, as well as some P and T axes derived from nodal planes or viceversa) (see table below)

| Source | day | month | year |
|---|---|---|---|
| Muller, 1977 | 06 | 05 | 1976 |
| Eva & Pastore, 1993 | 13 | 09 | 1989 |
| Poli et al., 2002 | 05 | 10 | 1991 |
| Bressan et al., 2018 | 01 | 06 | 2009 |
| Restivo et al., 2016 | 28 | 10 | 2010 |
| ISC | 02 | 02 | 2013 |
| Romano et al., 2019 | 02 | 05 | 2013 |
| Romano et al., 2019 | 12 | 05 | 2015 |
| Bressan et al., 2018 | 23 | 11 | 2017 |

Thanks for bringing this to our attention. We cross-checked the data and corrected the typos

19) I suggest to eliminate trend and plunge of B axis from the catalogue and to investigate about some solutions (table below) to improve the database.

We understand the reviewer concern. However, for the sake of completeness, we prefer to keep the B-values we used to calculate the Kagan angle. We have recalculated the Sugan 2020 B-values to bring them into agreement with the convention used by the other authors.

**Rev.2**

The goal of the paper is in line with ESSD journal. The methods applied in building the FPS database are standard. What the work brings new is the completing the existing FPS catalogues with new determinations, revising the previous solutions and merging and integrating all the available information with the new determinations. The data are complete and easy to access and have significant potential to be used in the future. Also the format of metadata is appropriate. The presentation is clear and concise except Introduction, which is too long in my opinion. The first part describes the methods used to compute FPS, but all this information is familiar for any interested reader. I recommend to shorten as much as possible this part.

We acknowledge the reviewer for the appreciation. We have reduced the introduction as suggested to improve the readability of our work.

Angela Saraò on behalf of the coauthors